# Changes in the Gut Microbiota Composition during Implantation of Composite Scaffolds Based on Poly(3-hydroxybutyrate) and Alginate on the Large-Intestine Wall

**DOI:** 10.3390/polym15173649

**Published:** 2023-09-04

**Authors:** Andrei A. Dudun, Dariana V. Chesnokova, Vera V. Voinova, Anton P. Bonartsev, Garina A. Bonartseva

**Affiliations:** 1A.N. Bach Institute of Biochemistry, Research Center of Biotechnology of the Russian Academy of Sciences, Leninsky Ave. 33, Bld. 2, 119071 Moscow, Russia; dudunandrey@mail.ru; 2Faculty of Biology, M.V. Lomonosov Moscow State University, Leninskie Gory 1-12, 119234 Moscow, Russia; daryana8@yandex.ru (D.V.C.); veravoinova@mail.ru (V.V.V.); ant_bonar@mail.ru (A.P.B.)

**Keywords:** biopolymers, poly(3-hydroxybutyrate), alginate, gut microbiota, probiotics, antibiotics, metagenomics, 16S analysis, bioinformatics

## Abstract

The development of biopolymer scaffolds for intestine regeneration is one of the most actively developing areas in tissue engineering. However, intestinal regenerative processes after scaffold implantation depend on the activity of the intestinal microbial community that is in close symbiosis with intestinal epithelial cells. In this work, we study the impact of different scaffolds based on biocompatible poly(3-hydroxybutyrate) (PHB) and alginate (ALG) as well as PHB/ALG scaffolds seeded with probiotic bacteria on the composition of gut microbiota of Wistar rats. Implantation of PHB/ALG scaffolds on the large-intestine wall to close its injury showed that alpha diversity of the gut microbiota was not reduced in rats implanted with different PHB/ALG scaffolds except for the PHB/ALG scaffolds with the inclusion of Lactobacillus spheres (PHB/ALG-L). The composition of the gut microbiota of rats implanted with PHB/ALG scaffolds with probiotic bacteria or in simultaneous use of an antimicrobial agent (PHB/ALG-AB) differed significantly from other experimental groups. All rats with implanted scaffolds demonstrated shifts in the composition of the gut microbiota by individual operational taxonomic units. The PHB/ALG-AB construct led to increased abundance of butyrate-producing bacteria: *Ileibacterium* sp. dominated in rats with implanted PHB/ALG-L and *Lactobacillus* sp. and *Bifidobacterium* sp. dominated in the control group. In addition, the PHB/ALG scaffolds had a favourable effect on the growth of commensal bacteria. Thus, the effect of implantation of the PHB/ALG scaffold compared to other scaffolds on the composition of the gut microbiota was closest to the control variant, which may demonstrate the biocompatibility of this device with the microbiota.

## 1. Introduction

The gut microbiota, as a single organ, plays a major role in human health. Thanks to the development of techniques in genomics, metagenomics, and metabolomics in recent years, it has become clear that the gut microbiota is responsible for many vital functions of the human body [1]. The influence on digestive, defensive, synthetic, immune, and regulatory functions shows that the entire gut microbiome is not just a passive observer, but one of the key organs in maintaining life [2]. Thus, a disruption in the homeostasis of the intestinal microbial community may lead to pathological changes in the organism. Recent studies have shown that in various gastrointestinal (GI) diseases, such as inflammatory bowel disease, multiple sepsis, ulcerative colitis, etc., the composition of the intestinal community is significantly altered compared to the gut microbiota of healthy individuals [3,4].

Sometimes, in cases of severe diseases of the large intestine, such as colorectal cancer, surgical intervention is required to resect the damaged part of the large intestine [5]. As clinical practice shows, quite often, surgical intervention on the large intestine is accompanied by complications in patients [6]. For example, in a study analyzing clinical cases of 600 patients undergoing colectomy, more than 25% had ulcerative colitis [7]. Also, the most common complications after colorectal resection are infections, wound infections, or infections of internal organs, as well as GI complications, including intestinal obstruction [6]. In this context, new approaches are needed for direct invasive treatment of the large intestine, one of which is the use of tissue-engineered 2D- and 3D-scaffolds based on biocompatible biopolymers. Currently, research on the tissue engineering of the mucosa and the entire intestinal wall has not yet received widespread attention as in the case of bone or nerve tissue [8]. However, certain developments have been made in this field during the last three decades. For example, various polymeric and cellular constructs have been obtained and used in GI surgery [9,10].

However, no study has yet shown how exactly one or the other tissue engineering approach affects the gut microbiota. Most research on gut tissue engineering has relied on histological and immunochemical methods [11]. Meanwhile, the dense symbiotic relationship of intestinal tissues with complex bacterial communities leads to significant changes of the microbiota composition in response to various physiological and pathological processes in the GI tract. Therefore, the composition of the microbiota can play the role of a marker during implantation of various medical devices and tissue-engineering scaffolds, and we can trace the possible roles and functions of certain taxonomic bacterial groups in the healing of damaged intestinal tissue. In this case, we can speak of “microbiota-associated biocompatibility” of the applied biomaterial. In contrast to the biocompatibility of biomaterials with different tissues and organs, in which parameters such as cell viability, adhesion, migration and proliferation in vitro, and tissue reaction and regeneration in vivo are primarily considered, the term “microbiota-associated biocompatibility” implies the restoration of the qualitative composition and the number of inhabitants of the bacterial community by introducing different additives such as prebiotics and probiotics or by implantation of various medical devices from different biomaterials [12,13]. Thus, the study of changes in bacterial communities during intestine wall regeneration after surgical intervention will allow us to create a predictive model for the use of different biomaterials in GI surgery.

In this study, we developed a composite biopolymer scaffold in the form of an intestinal patch based on bacterial polymers PHB and ALG to investigate the gut microbiota of rats after implantation of these constructs. To achieve this goal, the first objective was the synthesis of PHB and ALG by the bacterial strain *Azotobacter vinelandii* 12 and the subsequent isolation and purification of these biopolymers. Different variants of composite scaffolds were then developed based on the synthesized polymers. The next step was a series of operations to implant different scaffolds into the large intestine of Wistar rats. The final stage involved 16S sequencing and analysis of the qualitative and quantitative composition of the gut microbiota in the rats after surgery. This study provides the basis of the observation that the microbiota may be a diagnostic marker that can provide an accurate predictive result for the detection of inflammatory or regenerative processes in the implantation of natural polymer medical devices.

## 2. Materials and Methods

### 2.1. Synthesis of PHB and ALG

PHB and ALG were synthesized by the bacterial strain *Azotobacter vinelandii* 12, selected from the collection of the Laboratory of Biochemistry of Nitrogen Fixation and Nitrogen Metabolism of the Bach Institute of Biochemistry of the Federal Research Center “Fundamentals of Biotechnology” of the Russian Academy of Sciences (Moscow, Russia).

It was previously demonstrated in our studies that these biopolymers synthesized by bacteria of the genus *Azotobacter* sp. are biocompatible and biodegradable [14,15]. PHB is a semi-crystalline hydrophobic polyester of 3-hydroxybutyric acid, in which the crystalline phase dominates over the amorphous phase [16]. This polymer is a solid substance, which can be successfully used to create substrates or three-dimensional structures for regenerative surgery. ALG is a hydrophilic unbranched polysaccharide composed of mannuronic and guluronic acid residues [17]. The main feature of ALG is that it forms gels due to ionotropic interactions of calcium ions with guluronic residues [18].

Based on our earlier published work [19], the cultivation of *A. vinelandii* 12 for optimal synthesis of PHB with molecular mass in the range from 250 to 600 kDa was performed on liquid Burke’s medium in the presence of low concentrations of sucrose and phosphate, as well as a high level of aeration (210 rpm). Optimal synthesis of ALG was also performed when the bacteria were grown on liquid Burke’s medium but in the presence of high concentrations of phosphate [19]. Cultivation was carried out in 750 mL shake flasks (volume of nutrient medium 200 mL) at constant pH = 7.2 in the medium and at a temperature of 28 °C.

### 2.2. Isolation and Purification of PHB and ALG

The first step for isolation of PHB and ALG was the separation of cell biomass after cultivation and culture liquid by centrifugation at 11,000× *g* for 30 min. Next, the first polymer was used to isolate ALG from the cell biomass by adding 1M NaCl (Chimmed, Moscow, Russia) and 100 mM EDTA (Sigma-Aldrich, Darmstadt, Germany) solution to it in a 1:8:1 ratio. The resulting mixture was then incubated for 1 h at 60 °C with stirring on an orbital shaker (PSU-20i, Biosan, Riga, Latvia) until complete homogenization of the solution. The supernatant was then obtained by centrifugation at 11,000× *g* for 30 min. The supernatant was precipitated with 3 volumes of chilled 80% ethanol at 4400 g for 15 min; at the end, the obtained precipitate was lyophilized for 24 h. The final step for purification of alginate was to dissolve the precipitate in 1M NaCl solution and perform a dialysis on the resulting solution against 1 L of 0.1 M NaCl for 30 h. To obtain purified alginate, the supernatants after dialysis were again precipitated with 3 volumes of ethanol and the precipitates were lyophilized [19].

The isolation of PHB from cell biomass was carried out by chloroform extraction for 12 h at 37 °C. The obtained extract was separated from cell residues by filtration and then PHB was isolated from the chloroform extract by precipitation with isopropyl alcohol. The step of dissolution in chloroform and precipitation of PHB with isopropyl alcohol was repeated at least 3 times. PHB was dried at 60 °C [18].

### 2.3. Design of PHB/ALG Scaffolds

Porous constructions based on PHB and ALG have been developed. The first step for this was the preparation of porous polymer microspheres of PHB. The fabrication of microspheres with a diameter of 250 μm was carried out by double emulsification method. A 5% (*w*/*v*) aqueous solution of ammonium carbonate (Macklin, Shanghai, China) was used as a porogen. A chloroform solution of PHB (M_w_ = 3.5 × 10^5^ g/mol) at a concentration of 42 mg/mL was mixed with an aqueous solution of ammonium carbonate in a ratio of 35:11. From the resulting mixture, a finely dispersed colloid was prepared using an IKA T 25 digital Ultra-Turrax homogenizer (IKA-Werke, Staufen, Germany) for 5 min at 15,000 rpm. The resulting colloid was added dropwise to a 1% (*w*/*v*) aqueous solution of polyvinyl alcohol (PVA) (Mowiol® 8-88, Sigma Aldrich, Darmstadt, Germany) under constant stirring on a top-drive R2R 2021 stirrer (Heidolph, Schwabach, Germany) at 750 rpm. Stirring was carried out until complete evaporation of the chloroform from the particles. After evaporation of the chloroform, the particles were separated from the emulsifier PVA by sedimentation and washing three times in hot distilled water (70 °C). The hot water also allowed to eliminate the residual porogen inside the spheres. The quality of washing was checked using wet standard indicator paper (Chimmed, Moscow, Russia) by the absence of alkaline reaction.

The washed microspheres were dried from distilled water and sieved through laboratory sieves U1-ESL (Kraft, Chelyabinsk, Russia) with mesh cloth according to GOST 6613-86 with mesh sizes of 94 μm and 315 μm for additional more accurate separations of microspheres by their diameters. Thus, microspheres with an average diameter of 250 ± 60 μm were obtained, possessing connected porosity with an average pore diameter of 16 ± 6 μm. The particle and pore sizes were estimated from scanning electron microscope images using ImageJ software (version 1.54d (NIH, Bethesda, MD, USA)).

The next step was to prepare a 100 μm thick PHB film on the surface of which the obtained spheres would be immobilized. For this purpose, the chloroform solution of the polymer was placed on a degreased glass Petri dish and allowed to evaporate completely. The resulting film was then coated with a thin layer of the polymer solution and the surface of the film was covered with a uniform layer of spheres. The obtained structure was also evaluated by scanning electron microscope images. At the end, the obtained film with spheres was cut out in the form of a plate (patch) with a length of 15 mm and width of 5 mm. The finished products were additionally washed three times with distilled water to remove impurities and weakly attached spheres.

The final step of the design was to place a porous substrate with PHB microspheres in a 1% ALG solution. The 1% ALG solution itself was in four variants: ALG, ALG with an encapsulation of bacterial strain *Lactobacillus fermentum* 90 TS-4 (L), ALG with an encapsulation of bacterial strain *Bifidobacterium longum* MC-42 (B), and ALG with a co-encapsulation of bacterial strains *Lactobacillus fermentum* 90 TS-4 and *Bifidobacterium longum* MC-42 (LB); a visual design scheme of the developed composite PHB/ALG scaffolds is presented in Figure 1. Cultivation of probiotic bacteria is described in Appendix A. 

For this purpose, initially, ALG is sterilized by pasteurization before inoculation of probiotic bacteria; the preparation of 1% alginate solution is standard for encapsulation of various cells [20]. One gram of bacterial alginate was dissolved in 90 mL of MRS broth, followed by seeding of probiotic bacteria. Seeding was performed with 10 mL of bacteria grown on MRS broth to achieve 1% alginate concentration, and the resulting mixture was thoroughly stirred on a magnetic stirrer. The final step was to dropwise add the obtained solution into 50 mM CaCl_2_ (Chimmed, Moscow, Russia). Due to ionotropic binding of calcium ions with guluronic monomers, spheres filled with alginate gel and bacteria were formed. Afterwards, the spheres were washed with phosphate-buffered saline (PBS) solution (Sigma-Aldrich, Darmstadt, Germany). The fabricated porous microsphere PHB constructs were incubated for 1 h in different variants of alginate 1% solution (ALG, ALG-L, ALG-B, and ALG-LB), after which the biopolymer products were transferred into 10% CaCl_2_ solution and also incubated for 1 h to create ALG hydrogel on the surface of the PHB plate.

### 2.4. Animal Experiments

Male Wistar rats of ten weeks of age and weighing 300–350 g were used for a series of large intestine surgeries. Keeping the laboratory animals and all manipulations with them, the study was carried out according to the ISO 10993-1:2009 ethical guidelines and approved by the Institutional Ethics Committee of the Research Center of Biotechnology (protocol code N22-D dated 12 February 2020). The rats were controlled under standard pathogen-free conditions at 25 ± 3 °C, 55% humidity, with a 12 h light/dark cycle and with constant access to food and water. Several experimental groups were formed during the surgical procedures to implant the construct into the large intestine: a control group in which rats did not undergo surgery (control, n = 4), a negative control (fake) in which rats underwent surgery but without implantation of the PHB/ALG scaffold (fake, n = 4), PHB/ALG scaffold (PHB/ALG, n = 4), PHB/ALG scaffold where rats were intramuscularly injected once a day for a week with a solution of the antibiotic Azithromycin (Hemofarm, Vršac, Serbia) at a concentration of 30 mg/kg of rat weight (PHB/ALG-AB, n = 6), rats with implanted PHB/ALG scaffold seeded with *Lactobacillus fermentum* 90 TS-4 (PHB/ALG-L, n = 4), rats with implanted PHB/ALG scaffold seeded with *Bifidobacterium longum* MC-42 (PHB/ALG-B, n = 4), and a group of rats with implanted PHB/ALG scaffold seeded with *Lactobacillus fermentum* 90 TS-4 and *Bifidobacterium longum* MC-42 (PHB/ALG-LB, n = 4).

### 2.5. Surgeries

During the operations, with secure fixation, the large intestine was selected in the GI tract of the rat, for the reason that the blind intestine has an irregular “pouch-like” shape for attachment. Therefore, all further experiments were performed on the large intestine, which directly emerges from the cecum (Appendix A).

Before the series of surgeries, rats were injected with 10 mg of Zoletil 50 (Virbac, Carros, France) per half kilogram of animal weight in combination with the muscle myorelaxant Rometar (Bioveta, Ivanovice na Hané, Czech Republic) at a dose of 6 mg/kg. Afterwards, a series of microsurgical manipulations were performed to bring the large intestine to the outside, create a lesion in the large intestine, implant a PHB/ALG patch at the site of the defect and apply sutures to close the damaged area [21,22,23]. A detailed description of the series of surgeries for implantation of different variants of the scaffolds is presented in the Appendix A).

### 2.6. Genomic DNA Isolation

After 7 days, the animals were sacrificed by an overdose of anesthesia and relaparotomy was performed to obtain biological material. Sampling was performed at the surgical site using a medical spatula from the wall of the large intestine.

All obtained samples were stored in a freezer at −70 °C until DNA was isolated from the biomaterial. For DNA isolation, silicon-zirconium beads (BioSpec Products, Bartlesville, OK, USA) with a diameter of 0.1 mm (300 mg) and 0.5 mm (100 mg) and 1200 µL of warm lysis buffer (500 mM NaCl, 50 mM Tris-HCl, pH 8.0, 50 mM EDTA, 4% SDS) were used (all reagents were obtained from Sigma-Aldrich; Darmstadt, Germany). All this was mixed on a vortex and homogenized using a MiniBeadBeater (BioSpec Products, Bartlesville, OK, USA) for 3 min. The resulting lysate was then incubated at 70 °C for 15 min, after which the samples were centrifuged for 20 min at 14,000 rpm. The supernatant was then collected in new Eppendorfs and placed on ice. Lysis buffer was again added to the precipitate and the process was repeated. The supernatants were combined and 2 volumes of 96% alcohol (Chimmed, Moscow, Russia) and 1/10 volume of 3 M sodium acetate solution (Chimmed, Moscow, Russia) were added. The samples were then incubated at −20 °C for at least one hour. At the end of incubation, the samples were centrifuged at 12,000 rpm for 20 min. The resulting precipitate was washed twice with 80% ethanol (Chimmed, Moscow, Russia) and then air dried and dissolved in TE buffer (Evrogen, Moscow, Russia). The last step was the addition of RNase A (5 mg/mL) (Sigma-Aldrich, Darmstadt, Germany) to the eluates at a 1:200 ratio to the solution; this step is necessary to remove RNA molecules in the samples. The samples with RNase were incubated for 1 h at 37 °C and the resulting DNA solution was stored at −20 °C.

### 2.7. Sequencing of 16S rRNA Gene Amplicon

Metagenomic analysis to assess the qualitative and quantitative diversity of bacterial communities was performed using the hypervariable V4 region of the 16S rRNA gene. The sequence of the V4 site is 254 nucleotides and this DNA fragment in each sample was amplified using forward primer Forward515 with the sequence GTGBCAGCMGCCGCGGCGGTAA and reverse primer Reverse806 with the sequence GGACTACHVGGGTWTCTAAT.

Preparation of 16S DNA libraries was conducted using two-step PCR. At the second stage, Nextera XT Index Kit (Illumina, San Diego, CA, USA) was used to attach adapters and indexed barcodes according to the manufacturer’s protocol. After amplification, PCR products using different combinations of specific primers were purified using Agencourt AMPure XP beads (Beckman Coulter, Brea, CA, USA). The concentration of the resulting 16S libraries in solution was determined with a Qubit^®^ fluorimeter (Invitrogen, Waltham, MA, USA) using the Quant-iT™ dsDNA High-Sensitivity Assay Kit. Purified DNA libraries were mixed equimolarly according to the concentrations obtained. The quality of the library prepared for sequencing was assessed on a Bioanalyzer 2100 device (Agilent, Santa Clara, CA, USA) using the Agilent DNA 1000 Kit. The samples were then directly sequenced on a MiSeq instrument (Illumina, San Diego, CA, USA) using MiSeq Reagent Kit v2 reagents (300 cycles).

### 2.8. Bioinformatics Analysis

The sequencing data were analyzed using open-source bioinformatics software. First, all reads were quality trimmed and primer sequences were removed using the BBMerge application (version 38.90, https://github.com/BioInfoTools/BBMap/blob/master/sh/bbmerge.sh, accessed on 25 July 2023). Merging of overlapping paired-end reads was performed in the same software and then the assembled contigs in FASTA format were analyzed using the MOTHUR package v.1.45.0 [24,25]. During processing, reads corresponding to chloroplasts, mitochondria, or eukaryotic DNA were removed from the entire array. Taxonomy was also determined for all V4 sequences using the naive Bayesian classifier method in the QIIME 2 application in order to construct a phylogenetic tree. Visualization of the tree was performed in the iTOL v6 online application [26]. After processing the 16S data in the MOTHUR package, the results were imported into the R environment (version 3.6.3, Lucent Technologies, Murray Hill, NJ, USA, codenamed “Holding the Windsock” Copyright © 2020. The R Foundation for Statistical Computing). Visualization of the obtained results in R was performed using the libraries tidyverse, reshape, vegan, plyr, scales, ggcorrplot, devtools, ggbiplot, and ggplot2 [27,28,29].

### 2.9. Statistical Analysis

Statistical evaluation of data was performed using the tidyverse package in the R environment (version 3.6.3, Lucent Technologies, Murray Hill, NJ, USA, codenamed “Holding the Windsock” Copyright © 2020. The R Foundation for Statistical Computing). A non-parametric Kruskal–Wallis test was employed for all statistical analyses. Data were averaged with the standard error to the mean (± SD) and considered significant for *p* < 0.05.

## 3. Results

In summary, after a series of surgeries, a total of more than 60 fecal samples were collected from seven different groups of Wistar rats with different variants of implanted PHB/ALG biopolymer scaffolds: control group (11 samples), fake group (12 samples), PHB/ALG group (12 samples), PHB/ALG-AB group (15 samples), PHB/ALG-L group (four samples), PHB/ALG-B group (six samples), and PHB/ALG-LB group (four samples).

### 3.1. Characterisation of the PHB/ALG Scaffolds

An average diameter of the obtained microspheres was 250 ± 60 μm. As is shown in Figure 2, polymer microspheres have an interconnected porosity and can be impregnated with alginate with and without probiotic bacteria. An average pore diameter of the spheres was 16 ± 6 μm, which is enough for unhindered infiltration of bacterial cells into polymer spheres.

In Figure 3, the structure of the PHB construction before impregnation with alginate hydrogel with and without probiotic bacteria can be seen.

### 3.2. Analysis of the Gut Microbiota of Rats

Clustering of V4 sequences with a distance of less than 3% (97% identity or more) based on the SILVA 138.1 reference database yielded 86,594 operational taxonomic units (OTUs), of which 1044 occurred more than five times in the entire data set. The total V4 sequence data were presented as a phylogenetic tree.

The taxonomy of the phylogenetic tree was presented with a resolution down to the bacterial phylum (Figure 4). From the data obtained, a total of 19 phyla were identified. The dominant phylum in the gut microbiota of the studied Wistar rats was *Firmicutes*, which is fully consistent with the literature data [30,31].

Analysis of alpha diversity metrics by Shannon and Simpson indices showed low diversity of gut microbiota in rats with an implanted PHB/ALG scaffold and with ALG spheres of encapsulated Lactobacillus (PHB/ALG-L) (Figure 5A,B). Presumably, Lactobacillus-based spheres promoted aggressive growth of one single or several taxonomic groups, replacing many other microorganisms of the large intestine. It is possible that the metabolic products of the encapsulated bacteria gave a dramatic boost to the growth of a few taxonomic groups of bacteria. The so-called “cross-feeding” effect stimulates the growth of some microbes at the expense of metabolites from another bacterial group and sharply suppresses the growth of other bacterial groups [32]. It should be noted that the level of alpha diversity (except for PHB/ALG-L) was not reduced in different groups of rats compared to the control group (Figure 5A,B). This observation emphasizes that it is not always the case that a change in conditions, such as surgical intervention or artificial conditions, reduces the level of bacterial communities in the mammalian gut microbiota [33].

Analyses of the abundance profiles using non-metric multidimensional scaling (NMDS) and principal component analysis (PCA) techniques revealed an interesting feature (Figure 5C,D). The data demonstrate that the qualitative composition of the gut microbiota of the studied rats’ changes significantly, and two groups of samples can be distinguished according to the directionality of changes. The first group were rats that either did not have surgery or had surgery but without probiotic encapsulation or antibiotic therapy. The second group were rats with surgery and antibiotic or probiotic bacteria therapy. Thus, we can conclude that the effect of the antibiotic azithromycin and Lactobacillus and Bifidobacterium probiotics contributed more significantly to the qualitative composition of the intestinal community than the fact of the surgery itself. Indeed, many papers show that antibiotic or probiotic therapy shifts the gut microbiota towards one or the other taxonomic bacterial groups [36,37,38]. But there have been no previous comparisons of the effects of drugs during surgery.

Microbiota composition at the bacterial class level indicated two main dominant groups in all samples (Figure 6A).

Bacteria of the Clostridia and Bacilli classes had the main representation in all groups. It can be observed that the largest representation of Clostridia was in the group of rats with implantation of the biopolymer construct using an antibiotic (PHB/ALG-AB). This observation shows that most of the bacteria of the Clostridia class have resistance to this antimicrobial agent. This fact is very interesting because in the current view, it is generally accepted that Clostridia class bacteria such as *Clostridium difficile* are pathogens for humans and are responsible for many inflammatory and oncological diseases of the intestine [39,40]. In fact, most members of the Clostridia class in the gut microbiota play a commensal role and maintain homeostasis of the organism [41]. They have a protective function in the gut by synthesizing hydroxybutyric and other fatty acids, which in turn exhibit barrier and anti-inflammatory functions in the GI tract [42,43].

By increasing taxonomic resolution with genus accuracy, the SILVA database of V4 fragments of the ribosomal 16S RNA gene identifies a large number of OTUs for each group. Using the Kruskal–Wallis method of multiple comparisons, only 20 OTUs were statistically significant and nine of these had an accurately determined genus (Figure 6B).

The composition of the intestinal microbiota of the studied rats, including only statistically significant OTUs, showed that fecal samples collected from rats with the implanted construct and probiotic bacteria included (PHB/ALG-L, PHB/ALG-B, and PHB/ALG-LB) showed a predominance of bacteria of the genus *Ileibacterium* sp. (Figure 7A). Data on this genus are rather scarce in the literature; bacteria of this taxonomic group are primarily known for being responsible for the metabolism of mono- and disaccharides in a wide range with their degradation to acetates [44]. It is likely that this microbe is responsible for the low alpha diversity in the group of rat samples with ALG sphere-encapsulated *Lactobacillus* sp. As previously suggested in the alpha diversity results, *Ileibactreium* sp. Through cross-feeding utilized metabolites of encapsulated Lactobacillus as a substrate for active growth and division, thereby inhibiting the growth and division of other bacteria in the gut community [32]. The probiotic bacteria *Lactobacillus* sp. And *Bifidobacterium* sp. Have an increased representation in the group where rats underwent false surgeries (fake group). According to literature data, it is known that due to their symbiotic relationship with the host organism, probiotics actively promote healing of the GI tract, have anti-inflammatory effects, and improve intestinal peristalsis [45,46,47,48].

The results for groups of rats implanted with PHB/ALG scaffold without antimicrobial and probiotic agents showed increased numbers of bacteria of the genera *Allobaculum* and *Blautia*. (Figure 7A). Both organisms are typical representatives of the mammalian gut microbiota. *Allobaculum* sp. bacteria grow on organic substrates and are able to ferment glucose. The end products of these representatives are butyrate, lactate, and ethanol [49]. Little is known about bacteria of the genus *Blautia* sp., as many works are mainly devoted only to their isolation and full genome sequencing of various representatives of this group [50,51], but there are a number of studies that partially reveal their role in the gut microbiota. In one of these studies, *Blautia* sp. bacteria were shown to reduce the likelihood of lethality in acute graft-versus-host reactions, endowing these bacteria with a striking anti-inflammatory effect in the transplantation process without systemic suppression of host immunity [52]. The beneficial bacteria UCG-005 have an increased representation in the PHB/ALG group. This taxonomic group belongs to the family *Rumminococcaceae* and is one of the beneficial PHB-producing bacteria [53,54,55,56]. UCG-005–prevalent bacteria detected in our samples after 7-day antibiotic administration have anti-inflammatory effects against multiple chronic inflammatory diseases of the large intestine [43].

A closer look at Figure 7A reveals that it correlates well with the results for beta diversity (Figure 5C,D). Namely, that the relative abundance of all statistically significant OTUs can be divided into two groups. In the first goes the predominance of *Allobaculum* sp., *Bifidobacterium* sp., *Blautia* sp., *Collinsella* sp., *Holdemanella* sp., *Erysipelotrichaceae*, and *Lactobacillus* sp. bacteria in the gut microbiota in rats that did not undergo surgery or did but without encapsulating probiotics and a course of antibiotics. In the second group, namely rats with an implanted PHB/ALG scaffold and encapsulated probiotics or on antibiotic therapy showed increased numbers of butyrate-producing bacteria such as Clostridiaceae, Oscillospiraceae, and UCG-005 as well as bacteria of the genus *Ileibacterium* sp. Thus, by looking at taxonomically significant groups in detail, we can more clearly see the structural difference between the two large groups based on the beta diversity results.

Based on the Kruskal–Wallis multiple comparisons, Spearman’s rank correlation analysis was also calculated and shown (Figure 7B). The PHB/ALG-AB samples showed a positive correlation of PHB-producing bacteria of the Clostridia class, and at the same time, a negative correlation of potentially pathogenic bacteria of the Erysipelotrichaceae family. In contrast, the control and false surgery variants had a positive correlation with the growth of probiotics (*Bifidobacterium* sp. and *Lactobacillus* sp.) and a negative correlation in all bacteria belonging to the Clostridia class. These results clearly demonstrate that different approaches to the intestine wall healing related to specific taxonomic groups of the gut microbiota begin to predominate in such cases. Also noteworthy is the PHB/ALG-L rat group, where almost no OTUs (except for the positive correlation of bacteria of the genus *Ileibacterium* sp.) showed any correlation with respect to this group by Spearman rank analysis. These results further confirm that it was *Ileibacterium* sp. that occupied the entire niche in the gut microbiota, thereby reducing the alpha diversity, which may indicate a possible inflammatory or pathogenic process in the observed group.

## 4. Conclusions

The metagenomic results obtained show that the gut microbiota is a large community of microorganisms that is permeated by complex relationships between themselves and the host organism. The introduction of any object or the influence of any factor can dramatically change the structure of the intestinal community, as confirmed by our results, where each group of samples possessed a different bacterial composition. The results of 16S metagenomic profiling showed that despite implantation of different types of PHB/ALG scaffolds on the large intestine of Wistar rats, the level of alpha diversity according to the Shannon and Simpson indices did not decrease in comparison with the control group of rats. The only group, namely PHB/ALG-L, demonstrates a reduced level of alpha diversity in relation to the other studied groups. The taxonomic composition of the gut microbiota suggests that the increased abundance of bacteria of the genus *Ileibacterium* sp. contributed to the low alpha diversity, thus concluding that this bacterial taxonomic group has a pathogenic character and may be a probable marker of inflammatory processes. Using NMDS and PCA ordination methods, it was determined that all samples tested clustered into two large distinct structured groups: rats that did not undergo surgery or did but without the introduction of probiotic and antimicrobial agents; and the second group, rats implanted with PHB/ALG scaffolds with probiotics encapsulated in them or rats on a course of antibiotics. This result suggests that when using an intestinal patch based on our synthesized PHB and ALG, the composition of the intestinal microbial community was closer to the control, i.e., we can say that it is biocompatible with the gut microbiota. In contrast, the introduction of probiotic bacteria or a course of antibiotics has a pronounced effect on its composition. Data on the taxonomic composition of each individual group showed a major representation of two classes, namely Clostridia and Bacilli. The bacterial composition with resolution to genus had 20 statistically significant OTUs. From the relative abundance of 20 OTUs among all the studied groups, it is evident that these data correlate well with the beta diversity results. Thus, the groups of PHB/ALG-AB, PHB/ALG-L, PHB/ALG-B, and PHB/ALG-LB rats had increased numbers of butyrate-producing bacteria, while rats without surgery or falsely operated upon (fake) rats and rats with a PHB/ALG patch showed a predominance of bacteria *Allobaculum* sp., *Bifidobacterium* sp., *Blautia* sp., *Collinsella* sp., *Holdemanella* sp., *Lactobacillus* sp., and Erysipelotrichaceae family in the gut microbiota. Thus, it can be concluded that despite the surgical implantation of PHB/ALG composite constructs, the construct itself did not significantly alter the diversity of the gut microbiota according to the alpha diversity metric and the structure of the gut microbiota according to the beta diversity metric. These data suggest the potential “biocompatibility” of the PHB/ALG patch to the gut microbiota, which makes this construct promising for invasive treatment of severe colonic diseases in the future.

## Figures and Tables

**Figure 1 polymers-15-03649-f001:**
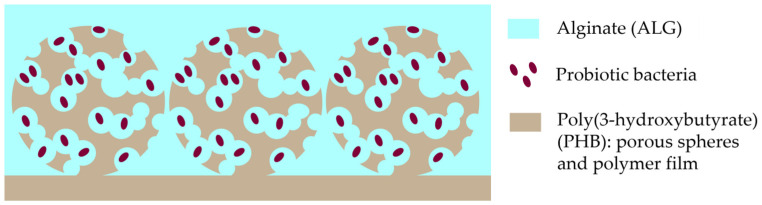
Scheme of the designed PHB/ALG construction.

**Figure 2 polymers-15-03649-f002:**
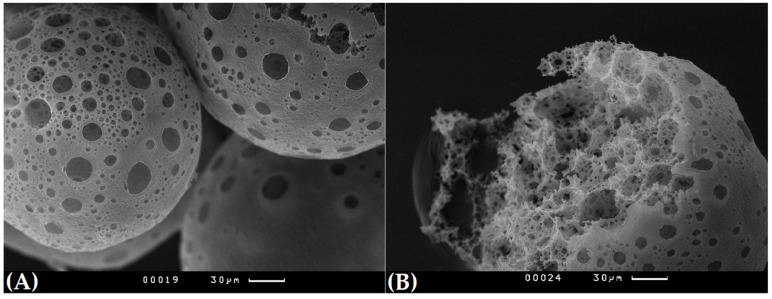
Porous surface of microspheres (**A**) and inner porosity (**B**) of microspheres. Scanning electron microscopy.

**Figure 3 polymers-15-03649-f003:**
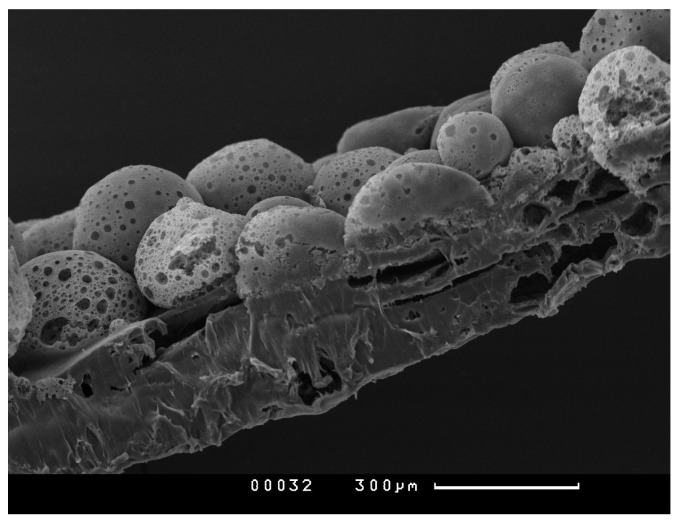
Porous spheres immobilized on a polymer film. Scanning electron microscopy.

**Figure 4 polymers-15-03649-f004:**
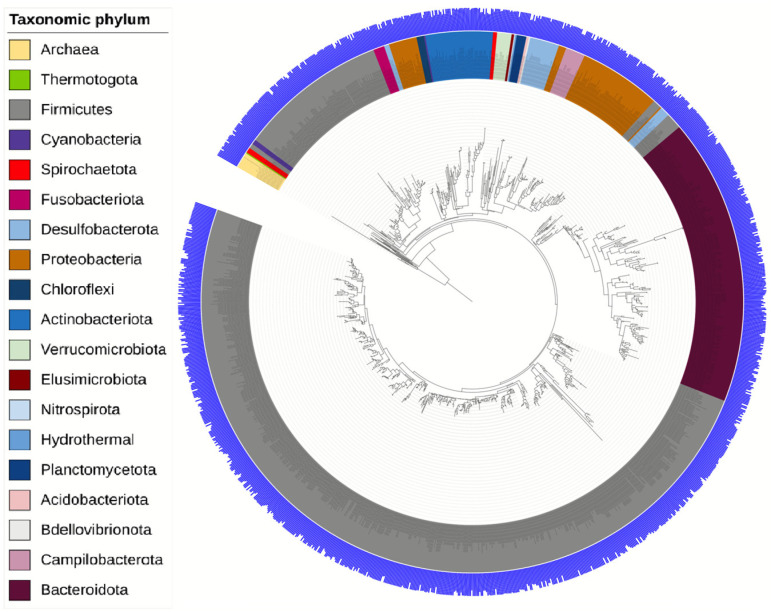
Rooted circular phylogenetic tree of all OTUs of the gut microbiota of Wistar rats. The tree was visualized in the iTOL online application.

**Figure 5 polymers-15-03649-f005:**
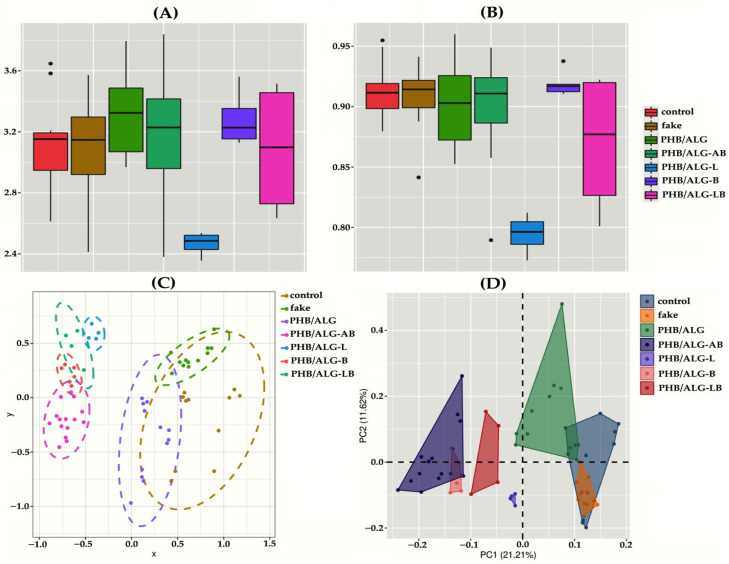
Analysis of gut microbiota diversity in different groups of rats. Graph visualization was created using R packages such as vegan [34] for calculations and analysis of diversity and differences and ggplot2 [35] for plotting. (**A**,**B**) Alpha diversity by Shannon index (**A**) and Simpson index (**B**). (**C**,**D**) Beta diversity analysis shown using ordination methods, namely (**C**) non-metric multidimensional scaling (NMDS) and (**D**) principal component analysis (PCA).

**Figure 6 polymers-15-03649-f006:**
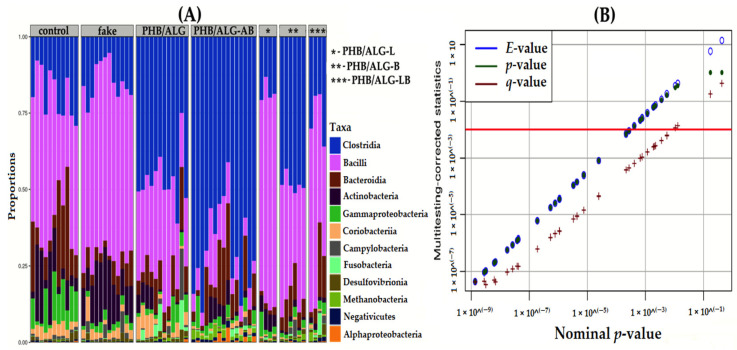
Microbial abundance analyses based on 16S rRNA gene sequence. (**A**)—Class-level abundance of bacterial communities of the gut microbiota of the studied rats. (**B**)—Non-parametric method of multiple comparisons using the Kruskal–Wallis test. Statistical significance of individual OTUs was determined by *E*-value, *p*-value, and *q*-value.

**Figure 7 polymers-15-03649-f007:**
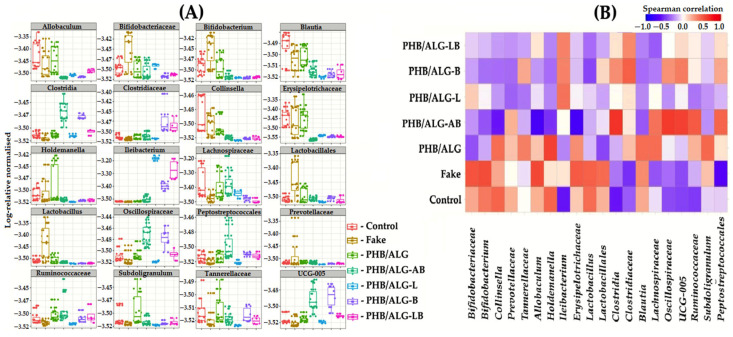
Microbial abundance analyses based on 16S rRNA gene sequence. (**A**)—Relative abundance of statistically significant OTUs between study groups. (**B**)—Spearman’s rank correlation analysis of statistically significant OTUs across study groups.

## Data Availability

The data presented in this study are available on request from the corresponding author. The data are not publicly available due to privacy.

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
