# Peer review of "Changes in the Gut Microbiota Composition during Implantation of Composite Scaffolds Based on Poly(3-hydroxybutyrate) and Alginate on the Large-Intestine Wall"

_polymers, 2023, doi:10.3390/polym15173649_

Round 1

Reviewer 1 Report

Review note

This paper designed a composite scaffolds based on poly(3-hydroxybutyrate) 3 and alginate and discovered changes in the gut microbiota composition following implantation. I appreciate the authors’ works. However, to grow into a publication, I think there are some issues the authors need to address.

1.     The introduction needs improvements. Usually, the introduction consists of several paragraphs, each of which briefly introduced the background and the gap of knowledge by the end. The last paragraph is the recap of the current study. The authors should exploit and point out the gap of knowledge of this study more clearly.

2.     The “fake” group is not well-defined. Quote “negative control (fake) or rats that underwent 183 surgery without implantation of the PHB/ALG scaffold (fake, n = 4)”, the authors mentioned two conditions using “fake”. How is “negative control” performed and how is it different from the blank control?

3.     The authors should explain the element in each panel clearly in both figure legends and in the related contents. For example, what are the dots indicating in Figure 5AB?

4.     Figure 6C is too small to read.

5.     Based on Figure2, the PHV/ALG scaffolds seemed to be more porous inside and relatively smooth on its surface? Does this design affect the probiotic bacteria in it interacting with the gut wall? The authors may try to modify the scaffolds achieving unanimous porosity and roughness.

In summary, I feel significant improvement is needed for this manuscript before it could be accepted. I hope the author(s) could find some of the above discussions helpful for improving the paper. 

Moderate editing of English language required

Reviewer 2 Report

Review Comments: In this study, researchers aimed to bridge health systems and polymers to contribute to the development of tissue engineering through biocompatible materials. The authors explored the impact of biopolymer scaffolds on the gut microbiota during intestine regeneration. This research opens up possibilities for using biopolymers in the treatment of colonic diseases. Additionally, the study showcases that biopolymers like poly(3-hydroxybutyrate) (PHB) and alginate (ALG) can be used for intestinal regeneration without altering the diversity and structure of the gut microbiota.

Top of Form

Review Comment: The topic of the manuscript is both original and relevant to the field as it offers sustainable and cost-effective treatment for colonic diseases using biocompatible biopolymers.

Review Comments: This study introduces the implantation of PHB/ALG biopolymer scaffolds in the intestines of Wistar rats and highlights the potential use of biomaterials to treat colonic diseases.

·         Novel Biomaterial: In this study, a novel biomaterial capable of implantation in the large intestine was introduced. This biomaterial possesses unique characteristics such as interconnected porosity and the ability to be impregnated with alginate and probiotic bacteria, providing advantages over other conventional biomaterials used in similar types of research.

·         Comprehensive Experimental Approach: The authors conducted a series of surgical and fecal analyses on different groups of rats, which, therefore, provides a more comprehensive and nuanced understanding of the potential effects on the gut microbiota.

·         Evaluation of Alpha and Beta Diversity: The evaluation of Alpha and Beta Diversity in this study provides insight into the structural changes in the gut microbiota induced by the implanted biomaterial.

Overall, the findings of this study are valuable contribution to the field of health, biomaterials and intestinal research.

 Review Comment: (i) The authors may propose a mechanism to elucidate how PHB/ALG biopolymer scaffolds interact with the gut microbiota. (ii) Perform the metabolomics analysis of the gut microbiota to provide understanding how biomaterials and probiotics influence the gut microbial composition and function. (iii) Additionally, conducting the study with rats having diseased intestines would add further value, as it could shed light on the potential therapeutic effects of the PHB/ALG biopolymer scaffolds in the context of colonic diseases.

Top of Form

Review Comment: The provided references for the study appear to be relevant and appropriate for the research topic.

Review Comment: It is suggested to create a colorful framework/workflow of the study to present the research in a single figure. This visual representation will help in clearly illustrating the different stages of the study, the experimental design, and the key findings, making it easier for readers to grasp the overall concept and flow of the research.

Figure 4 and Figure 6 a,c.  Please increase the font size of the legend.

Round 2

Reviewer 1 Report

The authors have addressed all of my concerns in the previous round of review. I don't have further questions regarding the latest version of the manuscript.